

# Species diversity and community structure of crustacean zooplankton in the highland small waterbodies in Northwest Yunnan, China

Xing Chen[1,2], Qinghua Cai[1], Lu Tan[1], Shuoran Liu[3], Wen Xiao[3] and Lin Ye[1]

[1] State Key Laboratory of Freshwater Ecology and Biotechnology, Institute of Hydrobiology, Chinese Academy of Sciences, Wuhan, Hubei, China
[2] University of Chinese Academy of Sciences, Beijing, China
[3] Institute of Eastern-Himalaya Biodiversity Research, Dali University, Dali, Yunnan, China

## ABSTRACT

Small waterbodies are a unique aquatic ecosystem with an increasing recognition for their important role in maintaining regional biodiversity and delivering ecosystem services. However, small waterbodies in Northwest Yunnan, one of the most concerned global biodiversity hot-spots, remain largely unknown. Here, we investigated the community structure of crustacean zooplankton and their relationships with limnological, morphometric and spatial variables in the highland small waterbodies in Northwest Yunnan in both the dry (October 2015) and rainy (June 2016) seasons. A total of 38 species of crustacean zooplankton were identified in our study, which is significantly higher than many other reported waterbodies in the Yunnan–Guizhou plateau as well as in the Yangtze River basin. This suggests that the highland small waterbodies are critical in maintaining regional zooplankton diversity in Northwest Yunnan. Meanwhile, we found limnological variables could explain most variation of crustacean zooplankton community, comparing to the morphometric and spatial variables in both the rainy and dry seasons. Our study revealed the diversity and community structure of crustacean zooplankton in the highland small waterbodies in Northwest Yunnan and highlighted the importance of small waterbodies in maintaining regional biodiversity.

## INTRODUCTION

Small waterbodies are critical for regional biodiversity and are increasingly recognized for their essential role in maintaining biodiversity and providing ecosystem services (*Williams et al., 2004*; *Biggs, von Fumetti & Kelly-Quinn, 2017*; *Kuczyńska-Kippen, 2020*). Small waterbodies with low density or without fish and abundant submerged vegetation support high biodiversity of aquatic organisms and contributed a large proportion of rare or endemic species to local freshwater habitats (*Williams et al., 2004*; *Oertli et al., 2005*; *Scheffer et al., 2006*). Also, small waterbodies have important ecological functions

Corresponding author
Lin Ye, yelin@ihb.ac.cn

(*Céréghino et al., 2014*; *Biggs, von Fumetti & Kelly-Quinn, 2017*). Small waterbodies can significantly reduce nutrient concentrations and protect downstream waters (*Cheng & Basu, 2017*). On the other hand, small waterbodies are vulnerable to environmental changes because of their small size (*Biggs, von Fumetti & Kelly-Quinn, 2017*).

Crustacean zooplankton is an important group in freshwater ecosystems because they occupy central positions in aquatic food webs, transferring energy to higher trophic levels (*Sommer et al., 1986*; *Fussmann, 1996*). In addition, crustacean zooplankton is sensitive to climate and environmental change (*Keller & Conlon, 1994*; *Shurin et al., 2010*; *Jones & Gilbert, 2016*). For quite a long time, the research on crustacean zooplankton in freshwater ecosystems has been mainly focused on lakes (*Barbiero et al., 2019*) and reservoirs (*Liu et al., 2020*). Yet, the ecology of crustacean zooplankton in highland small waterbodies remains seldom addressed.

Northwest Yunnan, located in Southwest China, has been designated as a global biodiversity "hot-spot" by World Wildlife Fund (WWF) and International Union for Conservation of Nature (IUCN) because of its rich biodiversity, unique and diverse highland landscape (*Mackinnon et al., 1996*; *Xu & Wilkes, 2004*; *Trizzino et al., 2014*). This region is in the upper stream of the Yangtze (Jinsha) River, the Mekong (Lancang) River, the Salween (Nujiang) River, and the Irrawaddy (Dulongjiang) River, attracting extensive attention of local and international communities (*Xu & Wilkes, 2004*; *Ao et al., 2021*). Currently, most ecology and biodiversity related studies in this region focus on the terrestrial vegetation and endangered wild animals (*Xu & Wilkes, 2004*; *Li et al., 2014*), yet still few studies addressed the aquatic ecosystems, especially for small waterbodies ecosystems.

In this study, we focus on the community structure and species diversity of crustacean zooplankton in highland small waterbodies in Northwest Yunnan, China. Besides the limnological variables (*e.g.*, water temperature, nutrients), previous studies have reported that morphometric variables (*e.g.*, surface area, depth) and spatial variables (*e.g.*, distance) also have critical effects on zooplankton diversity and community composition (*Dodson, 1992*; *Beisner et al., 2006*; *MacLeod, Keller & Paterson, 2018*). Here, we hypothesized that crustacean zooplankton in the small waterbodies are co-determined by limnological, morphometric, and spatial variables. Specifically, the main aims of our study are to understand: (i) the diversity and community structure of crustacean zooplankton in highland small waterbodies in Northwest Yunnan, (ii) the difference of community structure in rainy and dry seasons, (iii) how the limnological, morphometric and spatial variables determine the spatiotemporal variations of diversity and community structure.

## MATERIALS & METHODS

### Study sites and field sampling

The study sites were distributed on the east (Area E) and west (Area W) sides of a high mountain ridge 3,700 m in Gong-shan Country, Yunnan province, China (Fig. 1). The average annual temperature and precipitation (from August 1, 2014 to July 31 2015) was 7.7 °C and 2,515 mm respectively (*Liu et al., 2018*). There was a disused road lying across the "Area E", which separated this area into Upstream (EU) and Downstream (ED)

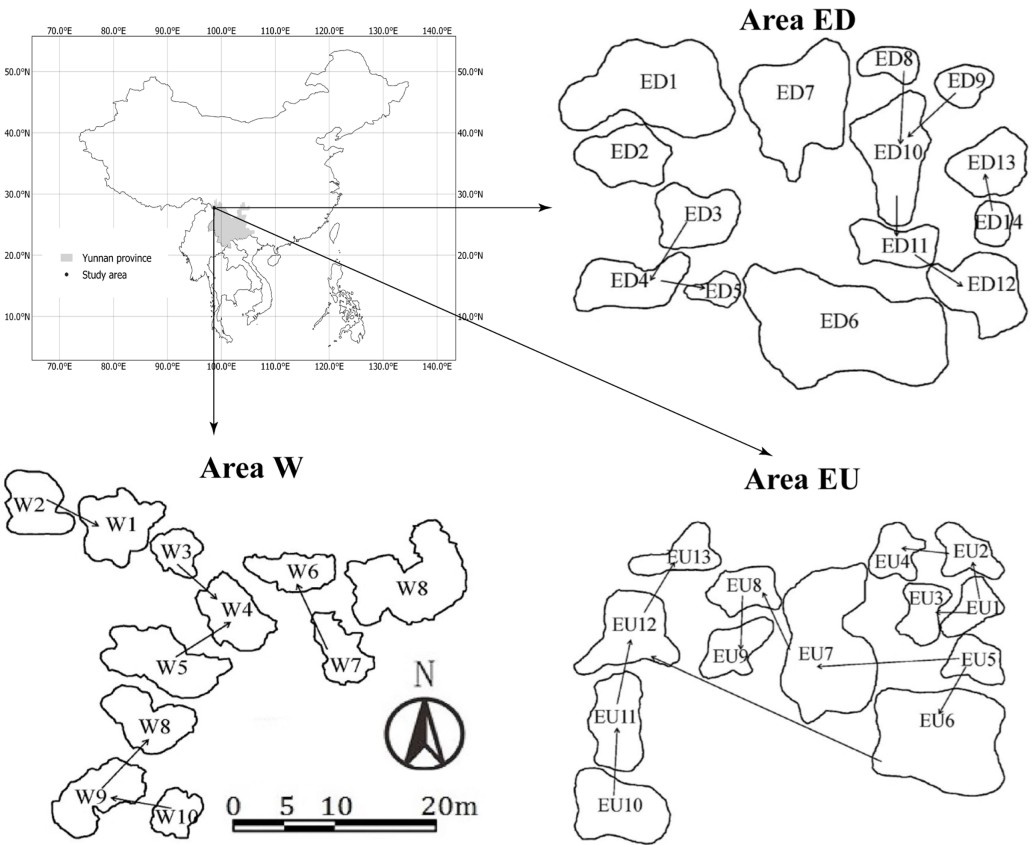

**Figure 1 Sampling areas and spatial distribution of small waterbodies in Gongshan County, Yunnan province, China.** The arrows represent the connectivity and water flow direction.

subgroups. The average elevation and area of small waterbodies are 3,131 m and 9.9 m$^2$ for the area W, 3,328 m and 13 m$^2$ for the area EU, and 3,274 m and 41 m$^2$ for the area ED, respectively. In addition, these small waterbodies have no fish, but have abundant macrophytes.

A total of two samplings were carried out in the dry (October 2015) and rainy (June 2016) seasons reflecting different hydrological regimes. A total of 30 and 32 small waterbodies were sampled in the dry and rainy seasons, respectively. For each small waterbody, the morphometric and spatial variables, including the water depth, water surface area, coordinates and altitude (using Garmin eTerx20, China) were measured. The physical parameters, including conductivity (Cond), dissolved oxygen (DO), pH, and water temperature (WT), were measured by a portable multi-parameter device (YSI Professional plus, Yellow Springs, OH, USA). Water samples for chemical analysis were collected from the center of each small waterbodies using a 350 ml plastic bottle. Ammonia nitrogen (NH$_3$N), nitrate nitrogen (NO$_3$N), total nitrogen (TN), phosphate (PO$_4$P), total phosphorus (TP), dissolved silicate (DSi), and dissolved organic carbon (DOC) were analyzed by segmented flow analyzer (Skalar SAN++; Breda, Netherlands), according to the user manual. Also, another 350 ml water sample was filtered through a micro-filter

(~1.2 µm, GF/C Whatman, Maidstone, UK) for the measurement of chlorophyll a (Chl-a). The concentration of Chl-a was measured with a spectrophotometer (Shimadzu UV-1800, Kyoto, Japan) with the standard method of *APHA (1999)*.

Crustacean zooplankton samples were collected with a plankton net (64 µm in mesh size) by filtering 20 L water sampled from the open water region in each small waterbody. All crustacean zooplankton samples were preserved with 5% formalin immediately.

## Zooplankton counting and identification

Crustacean zooplankton was counted and identified under the stereoscope (Zeiss Stereo Discovery V20, Oberkochen, German). All crustacean zooplankton samples were identified to the species level as far as possible. Specifically, all samples were screened under the stereoscope because of the low density of the crustacean zooplankters. The major reference books for identification were *Chiang & Du (1979)*, *Shen (1979)* and *Błędzki and Rybak (2016)*.

## Statistical analysis

A rarefaction was used to compare species richness and Shannon diversity between the rainy season and dry season because biodiversity was affected by sampling efforts, such as the number of sites and individual numbers (*Chao et al., 2014*). Specifically, we calculated species richness and Shannon diversity index for the whole waterbodies (*Chao et al., 2014*). Then, we plotted individual-based rarefaction curves for each season to compare the differences of species richness and Shannon diversity index.

A nonmetric multidimensional scaling (NMDS) was carried out to illustrate taxonomic and abundance similarity between the rainy and dry seasons. Further, the similarity percentage analysis (SIMPER) was conducted to investigate differences in community composition between the rainy and dry seasons and to determine the contribution of each species to the Bray–Curtis dissimilarities (*Clarke, 1993*).

In order to test our hypothesis, we conducted the variation partitioning with redundancy analysis (RDA) to compare species composition variation with the limnological, morphometric, and spatial variables. To avoid collinearity, only limnological variables with the correlation coefficient below 0.7 were selected as predictor variables (*Dormann et al., 2013*). As a result, the limnological variables, including TN, $NO_3N$, $NH_3N$, $PO_4P$, DSi, DOC, Cond, WT, Chl-a, were kept for further RDA. Water depth and surface area were selected as morphometric variables. Spatial variables can reflect the community dispersal limitation according to the metacommunity theory (*Heino et al., 2017*). The candidate spatial variables for the RDA were determined by Moran Eigenvector Maps (MEMs) (*Borcard & Legendre, 2002*). First, longitude and latitude were converted into Cartesian coordinates (the unit is kilometer). Second, the Euclidian distance matrix among the small waterbodies was calculated. Then, five eigenvectors with positive eigenvalues in MEMs were determined as the spatial predictors for RDA.

In the RDA, forward selection method was used to select the key variables explaining the variation of the crustacean zooplankton community (*Blanchet, Legendre & Borcard, 2008*).

To reduce the weight of species abundance, abundance data were Hellinger transformed before variation partitioning (*Legendre & Gallagher, 2001*). Finally, five limnological variables (NO$_3$N, DSi, Cond, WT and DO), two morphometric variables (water depth and surface area) and four spatial variables (MEM1, MEM2, MEM3 and MEM5) were selected in variation partitioning (Table S2). All analyses were implemented with R statistical software (*R Development Core Team, 2020*). Rarefaction was carried out with "*iNEXT*" package (*Hsieh, Ma & Chao, 2016*). MEMs and RDA variation partitioning were performed using "*vegan*" package (*Oksanen et al., 2019*).

## RESULTS

### Community composition

A total of 38 crustacean zooplankton taxa, including 20 Cladocera and 18 Copepoda species, were identified (Table 1). In the rainy season, the most common species were *Cyclops vicinus*, *Mesocyclops leuckarti*, *Alona affinis*, *Microclops varicaricans*, *Moina irrasa*, *Cyclops strenuuss*, *Ectocyclops phaleratus*, which occurred in more than 50% of the surveyed small waterbodies. In the dry season, *Chydorus ovalis*, *M. varicaricans*, *Tropocyclops prasinus*, *Ceriodaphnia laticaudata*, *Alonella exigua*, had a relative occurrence above 50% (Table 1).

The species accumulation curves showed that we have sampled considerable individuals in both the rainy and dry seasons (Fig. 2). The observed species richness is almost same as the estimated values of species richness in both the rainy and dry seasons. And the species richness in the dry season is significantly higher than that in the rainy season (Fig. 2A). However, Shannon diversity index showed that an explicit overlapping of observed and estimated species richness for the rainy and dry seasons (Fig. 2B).

The composition and abundance of crustacean zooplankton changed significantly between the rainy and dry seasons., *M. varicaricans*, *C. ovalis*, *C. vicinus*, *A. exigua* and *S. sarsi* are most influential species based on cumulative contribution (Table 2). Further, species compositions differed significantly between the rainy and dry seasons (Fig. 3).

### Crustacean zooplankton community variation partitioning

Limnological variables explained the most variation of crustacean zooplankton community in both the rainy (NO$_3$N, DSi, Cond and DO) and dry (NO$_3$N and WT) seasons, compared to the morphometric and spatial variables (Fig. 4). In the dry season, the limnological, morphometric, and spatial variables explained 23.69% of the crustacean zooplankton community structure (Fig. 4A). The limnological variables explained the most variation of zooplankton community structure (7.01%), which is significantly higher than spatial variables (3.44%) and morphometric variables (1.70%). Variation partitioning revealed 7.31% of the shared variation between limnological variables and spatial variables. However, only 1.48% of the variation was shared between the morphometric and spatial variables.

In the rainy season, all predictors explained 26.65% of the crustacean zooplankton community structure (Fig. 4B), which was slightly higher than the dry season.

**Table 1 Relative occurrences of crustacean zooplankton species in all samples, samples in area E, and samples in area W in the rainy (32 samples) and dry (30 samples) seasons.**

| Species | Rainy season | | | Dry season | | |
|---|---|---|---|---|---|---|
| | % of all samples | % of E samples | % of W samples | % of all samples | % of E samples | % of W samples |
| *Alona affinis* | 65.6 | 72.7 | 50.0 | 0.0 | 0.0 | 0.0 |
| *Moina irrasa* | 56.3 | 54.5 | 60.0 | 0.0 | 0.0 | 0.0 |
| *Chydorus ovalis* | 40.6 | 31.8 | 60.0 | 93.3 | 90.1 | 100.0 |
| *Diaphanosoma sp.* | 31.3 | 31.8 | 30.0 | 12.5 | 9.9 | 0.0 |
| *Bosmina coregoni* | 21.9 | 27.3 | 10.0 | 3.3 | 4.5 | 0.0 |
| *Alona guttata* | 12.5 | 18.2 | 0.0 | 3.3 | 4.5 | 0.0 |
| *Ceriodaphnia laticaudata* | 0.0 | 0.0 | 0.0 | 53.3 | 45.5 | 75.0 |
| *Alonella exigua* | 0.0 | 0.0 | 0.0 | 53.3 | 59.1 | 37.5 |
| *Alona karua* | 0.0 | 0.0 | 0.0 | 30.0 | 31.8 | 25.0 |
| *Graptoleberis testudinaria* | 0.0 | 0.0 | 0.0 | 20.0 | 18.2 | 25.0 |
| *Alona rectangula* | 0.0 | 0.0 | 0.0 | 23.3 | 22.7 | 25.0 |
| *Moina rectirostris* | 0.0 | 0.0 | 0.0 | 16.7 | 13.6 | 25.0 |
| *Ceriodaphnia quadrangula* | 0.0 | 0.0 | 0.0 | 20.0 | 9.1 | 50.0 |
| *Alonella globulosa* | 0.0 | 0.0 | 0.0 | 16.7 | 13.6 | 25.0 |
| *Ceriodaphnia reticulata* | 0.0 | 0.0 | 0.0 | 13.3 | 18.2 | 0.0 |
| *Alona quadrangularis* | 0.0 | 0.0 | 0.0 | 10.0 | 13.6 | 0.0 |
| *Chydorus barroisi* | 0.0 | 0.0 | 0.0 | 6.7 | 9.1 | 0.0 |
| *Alonella sp.* | 0.0 | 0.0 | 0.0 | 6.7 | 9.1 | 0.0 |
| *Alona sp.* | 0.0 | 0.0 | 0.0 | 3.3 | 0.0 | 12.5 |
| *Alonella nana* | 0.0 | 0.0 | 0.0 | 3.3 | 0.0 | 12.5 |
| *Cyclops vicinus* | 71.2 | 63.6 | 90.0 | 0.0 | 0.0 | 0.0 |
| *Mesocyclops leuckarti* | 71.2 | 63.6 | 90.0 | 0.0 | 0.0 | 0.0 |
| *Microclops varicaricans* | 62.5 | 50.0 | 90.0 | 90.0 | 86.4 | 100.0 |
| *Ectocyclops phaleratus* | 59.4 | 38.5 | 90.0 | 6.7 | 9.1 | 0.0 |
| *Cyclops strenuuss* | 56.3 | 45.5 | 80.0 | 3.3 | 0.0 | 12.5 |
| *Limnoithona sinensis* | 46.7 | 36.4 | 80.0 | 26.7 | 4.5 | 87.5 |
| *Nitocra lacustri* | 43.8 | 54.5 | 20.0 | 0 | 0 | 0 |
| *Sinodiaptomus sarsi* | 43.8 | 31.8 | 70.0 | 0 | 0 | 0 |
| *Eucyclops serrulatus* | 37.5 | 40.9 | 30.0 | 16.7 | 22.7 | 0.0 |
| *Sinocalanus dorrii* | 21.9 | 13.6 | 40.0 | 10.0 | 4.5 | 25.0 |
| *Onychocamptus mohammed* | 21.9 | 27.3 | 10.0 | 46.7 | 27.3 | 100.0 |
| *Neutrodiaptomus mariadvigae* | 15.6 | 13.6 | 20.0 | 10.0 | 0.0 | 37.5 |
| *Bryocamptus sp.* | 9.4 | 13.6 | 0.0 | 3.3 | 4.5 | 0.0 |
| *Tropodiaptomus hebereri* | 6.3 | 9.1 | 0.0 | 13.3 | 0.0 | 50.0 |
| *Tropocyclops prasinus* | 0.0 | 0.0 | 0.0 | 66.7 | 63.6 | 75.0 |
| *Paracyclops fimbriatus* | 0.0 | 0.0 | 0.0 | 16.7 | 18.2 | 12.5 |
| *Paracyclops affinis* | 0.0 | 0.0 | 0.0 | 10.0 | 9.1 | 12.5 |
| *Schmackeria inopinus* | 0.0 | 0.0 | 0.0 | 10.0 | 4.5 | 25.0 |

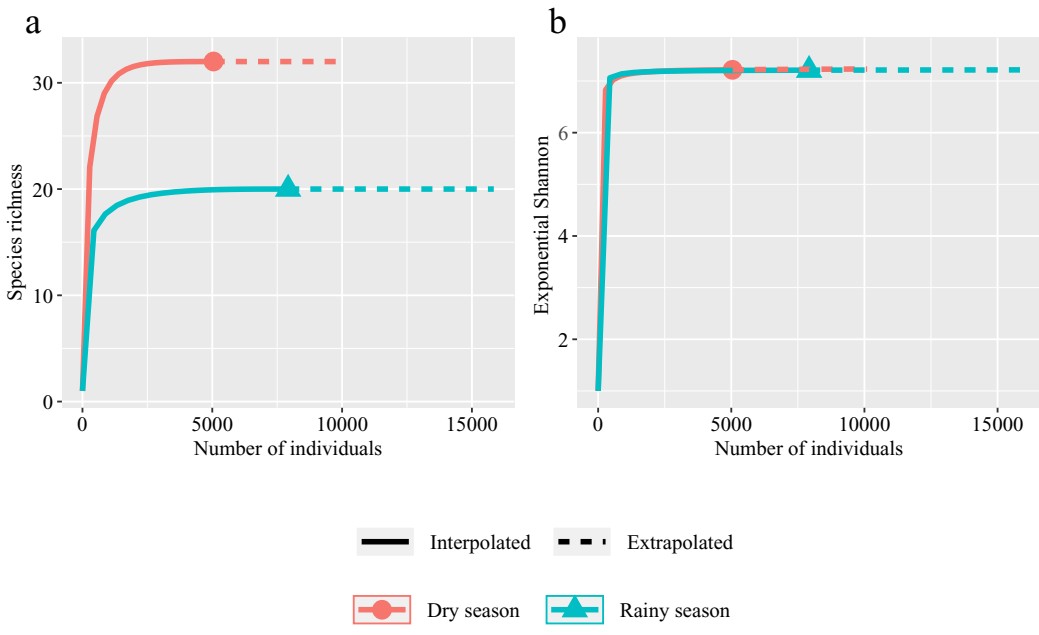

**Figure 2 Individual-based rarefaction for the dry (red) and rainy (green) seasons.** Symbols represent species richness (A) and Exponential Shannon (B). Continuous lines refer to interpolation, dotted lines refer to extrapolation.

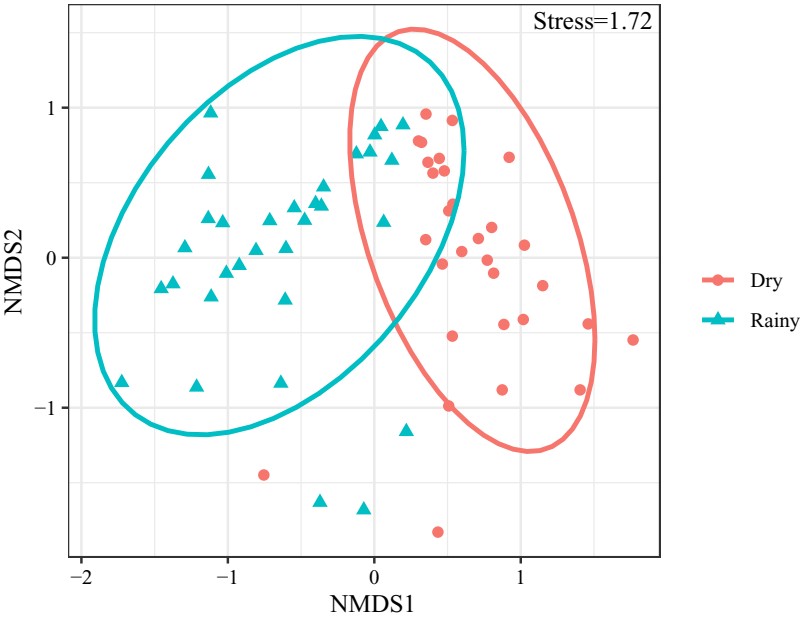

**Figure 3 Non-metric multidimensional scaling ordination (NMDS) of crustacean zooplankton communities.**

The limnological variables alone explained 18.12% of the variation. And the spatial variables had a lower contribution (3.45%) and followed by morphometric variables (0.64%).

**Table 2 Results of SIMPER analysis for species that accounted for the 90% of cumulative contribution.**

| Species | Average dissimilarity | Standard deviation | Ratio | Cumulative contribution | P |
|---|---|---|---|---|---|
| *M. leuckarti* | 0.20879 | 0.20616 | 1.0127 | 0.2313 | 0.675 |
| *M. varicaricans* | 0.15750 | 0.16750 | 0.9403 | 0.4058 | **0.001** |
| *C. ovalis* | 0.10450 | 0.14320 | 0.7297 | 0.5216 | **0.001** |
| *C. vicinus* | 0.07331 | 0.09875 | 0.7424 | 0.6028 | **0.001** |
| *A. exigua* | 0.06395 | 0.14393 | 0.4443 | 0.6737 | **0.019** |
| *S. sarsi* | 0.04930 | 0.12002 | 0.4108 | 0.7283 | 0.091 |
| *E. phaleratus* | 0.02948 | 0.04346 | 0.6783 | 0.7610 | **0.006** |
| *C. laticaudata* | 0.02447 | 0.05694 | 0.4299 | 0.7881 | **0.016** |
| *L. sinensis* | 0.02324 | 0.04526 | 0.5136 | 0.8139 | 0.600 |
| *C. strenuuss* | 0.02290 | 0.04765 | 0.4806 | 0.8392 | **0.034** |
| *M. irrasa* | 0.01874 | 0.03545 | 0.5286 | 0.8600 | **0.008** |
| *E. serrulatus* | 0.01756 | 0.04665 | 0.3765 | 0.8794 | 0.464 |
| *A. affinis* | 0.01276 | 0.01787 | 0.7145 | 0.8936 | **0.001** |

**Note:**
Bold values indicate statistical significance at the *p* < 0.05 level.

## DISCUSSION

One interesting finding of our study is that the species richness in our study area is significantly higher than many other reported waterbodies in the Yunnan–Guizhou plateau as well as in the Yangtze River basin (Table 3). For example, *Guo et al. (2009)* identified 36 crustacean zooplankton species in 13 different lakes in the Yunnan–Guizhou plateau with areas ranged from 10.7 to 297.9 km². Another similar research carried out in the plateau lake (Erhai Lake) in Yunnan province only recorded 11 crustacean zooplankton species for 12 field stations with 1-year continuous monthly monitoring (*Yang et al., 2014*). Comparing to the lakes and other waterbodies, small waterbodies usually have a high habitat heterogeneity which can support more diverse species and maintain a high diversity community (*Williams et al., 2004*).

The absence of predatory fish and complex habitat with abundant macrophytes might explain high crustacean zooplankton diversity in the highland small waterbodies in our study. Fish is more likely to be absent in small and isolated waterbodies because of high risks of extinction and low chances of colonization (*Scheffer et al., 2006*). In our field survey, we did not observe fish in any waterbodies. Presence of fish could profoundly impact crustacean zooplankton community structure by reducing species richness and simplifying community composition, especially in small waterbodies (*Scheffer et al., 2006*). The predation from fish is an important factor affecting crustacean zooplankton in small lakes (*Pinel-Alloul & Mimouni, 2013*). Meanwhile, some studies also suggested that macrophyte cover is important to maintain zooplankton diversity because of macrophyte provide good habitats for zooplankton (*Celewicz-Goldyn & Kuczynska-Kippen, 2017*). These natural, temporal, and mountain small waterbodies have good water quality and high coverage of macrophytes (*Kuczyńska-Kippen, 2020*), providing ecological niches for
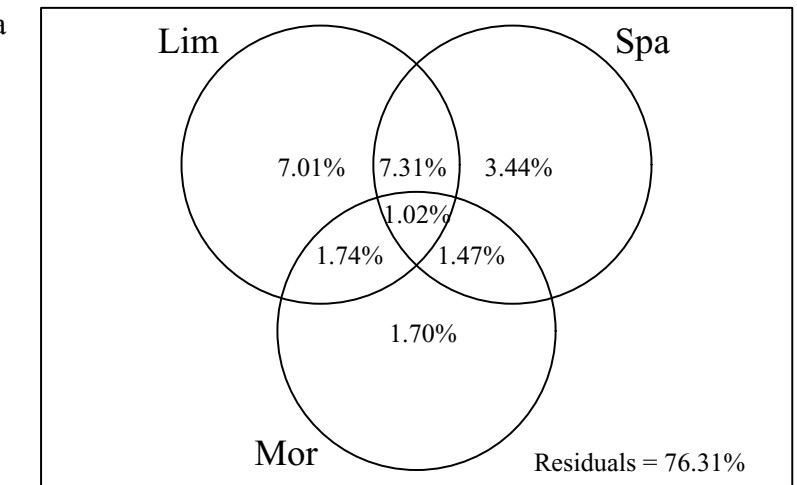

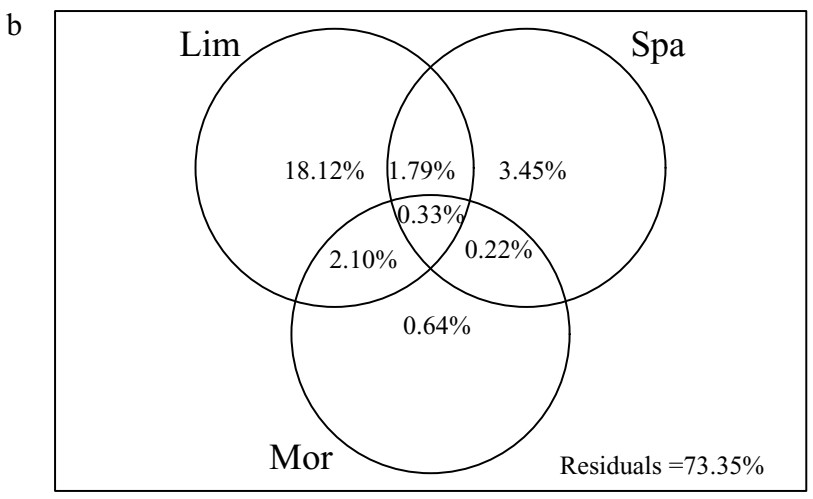

Values <0.001% not shown

**Figure 4 Venn diagram representing the variation partitioning of crustacean zooplankton community composition explained by explanatory variable.** (Lim) Limnological variables. (Spa) spatial variables represented by principal coordinates of neighbour matrices. (Mor) morphometric variables. (A) Dry season. (B) Rainy season.

rare (such as *Graptoleberis testudinaria* and *A. karua*) or endemic species (such as *T. hebereri* and *N. mariadvigae*).

The species compositions in the rainy and dry seasons are quite different in the highland small waterbodies in Northwest Yunnan, China. We found crustacean zooplankton richness was higher in the dry season compared to the rainy season. Higher richness in the dry season has also been reported in other studies and was associated with higher environmental heterogeneity and longer water residence time (*Pourriot, Rougier & Miquelis, 1997*; Melo & Medeiros, 2013), allowing more species to colonize in the small waterbodies. In terms of the species composition, *C. ovalis, M. varicaricans, T. prasinus, C. laticaudata, A. exigua* were the most common species in the dry season. However, in the rainy season, the common species shifted to *C. vicinus, M. leuckarti, A. affinis,*

**Table 3  A comparison of the species richness in the small waterbodies in the highland small waterbodies in Northwest Yunnan with other waterbodies in Yunnan-Guizhou plateau and Yangtze River basin.**

| Study area | Province | Area (km²) | Species richness | Reference |
|---|---|---|---|---|
| Thirteen lakes in Yunnan and Guizhou (*n* = 112) | Yunnan and Guizhou | 10.7~297.9 | 36 | *Guo et al., 2009* |
| Gaoyou Lake (*n* = 26) | Jiangsu | 674 | 26 | *Wei et al., 2017* |
| Chaohu Lake (*n* = 228) | Anhui | 780 | 23 | *Deng et al., 2008* |
| Lugu Lake (*n* = 36) | Yunnan | 57.7 | 23 | *Dong & Wang, 2014* |
| Fuxian Lake (*n* = 220) | Yunnan | 211 | 8 | *Pan et al., 2009* |
| Erhai (*n* = 144) | Yunnan | 249 | 11 | *Yang et al., 2014* |
| Our study (*n* = 62) | Yunnan | <0.001 | 38 | Our study |

**Note:**
   *n* indicating the number of total samples in the reported case.

*M. varicaricans*, *M. irrasa*, *C. strenuuss*, *E. phaleratus*. Among these species, we found two endemic species (*Tropodiaptomus hebereri* and *Neutrodiaptomus mariadvigae*) in the Yunnan–Guizhou plateau (*Shen, 1979*). Also, we found nine common species (*e.g.*, *C. vicinus*, *M. leuckarti*, *A. affinis*, *M. irrasa*) in the Yangtze River basin (*Chiang & Du, 1979*).

Our study also showed limnological variables explained most variation of crustacean zooplankton community in both the rainy and dry seasons, compared to the morphometric and spatial variables. This result is coherent with many other studies which also showed limnological variables as the most important factors in explaining variations of crustacean zooplankton compared to spatial variables. (*Heino et al., 2017*; *Lévesque et al., 2017*; *Brasil et al., 2020*). Our finding suggests that environmental filter played a key role in community structure in the highland small waterbodies in Northwest Yunnan, possibly related to their environmental heterogeneity. Previous experience showed that the environmental heterogeneity of small waterbodies in the Northwest of Yunnan depended on the watershed and precipitation (*Liu et al., 2018*).

We should add a caveat that not all potential limnological variables affecting the crustacean zooplankton communities were examined in our study due to limited data. Some researches suggested that macrophytes cover is important to maintain zooplankton diversity because macrophyte provide shelter from predators (*Cazzanelli, Warming & Christoffersen, 2008*; *Sagrario et al., 2009*). In our study, we did not address the effects of macrophytes. However, the zooplankton samples were collected in the open water area with no macrophytes, suggesting the direct effect of macrophytes on zooplankton samples was weak. Future works on factor shaping zooplankton community in small waterbodies could focus on the effect of macrophyte, which are probably important to affect zooplankton species assemblages (*Celewicz-Goldyn & Kuczynska-Kippen, 2017*).

## CONCLUSIONS

In this study, we reported the crustacean zooplankton community and their relationships with the limnological, morphometric and spatial variables in the highland small waterbodies in Northwest Yunnan for both the rainy and dry seasons. We identified 38 species of crustacean zooplankton, which is significantly higher than many other

waterbodies in the Yunnan–Guizhou plateau as well as in the Yangtze River basin. This suggests that small waterbodies are biodiversity hotspot and are important in maintaining regional zooplankton diversity in Northwest Yunnan. Limnological variables could explain the most variation of crustacean zooplankton community, comparing to morphometric and spatial variables in both the rainy and dry seasons. This study improved our understanding of the diversity and community structure of crustacean zooplankton in the highland small waterbodies in Northwest Yunnan and highlighted the importance of small waterbodies for biodiversity conservation and research.

## ACKNOWLEDGEMENTS

We thank Jun Sun, Xiaoyang He and Wenshu Yang for their assistance during field samplings.

### Funding

This work was supported by the National Natural Science Foundation of China (No. U1602262, 31670534, 31760126). The funders had no role in study design, data collection and analysis, decision to publish, or preparation of the manuscript.

### Grant Disclosures

The following grant information was disclosed by the authors:
National Natural Science Foundation of China: U1602262, 31670534, 31760126.

### Competing Interests

The authors declare that they have no competing interests.

### Author Contributions

- Xing Chen performed the experiments, analyzed the data, prepared figures and/or tables, authored or reviewed drafts of the paper, and approved the final draft.
- Qinghua Cai conceived and designed the experiments, authored or reviewed drafts of the paper, and approved the final draft.
- Lu Tan performed the experiments, prepared figures and/or tables, and approved the final draft.
- Shuoran Liu conceived and designed the experiments, performed the experiments, prepared figures and/or tables, and approved the final draft.
- Wen Xiao conceived and designed the experiments, authored or reviewed drafts of the paper, and approved the final draft.
- Lin Ye conceived and designed the experiments, performed the experiments, analyzed the data, prepared figures and/or tables, authored or reviewed drafts of the paper, and approved the final draft.

## Data Availability

The raw data are available in the Supplemental Files.

## Supplemental Information

Supplemental information for this article can be found online at http://dx.doi.org/10.7717/peerj.12103#supplemental-information.

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
