# Peer review of "Species diversity and community structure of crustacean zooplankton in the highland small waterbodies in Northwest Yunnan, China"

_PeerJ, doi:10.7717/peerj.12103_

## Round 0.1 · original submission · Major Revisions

While both reviewers saw merit in your work, they also pointed out a number of aspects that must be incorporated in the ms.

I particularly second R#1 when (s)he mentions that the ordination analysis must incorporate spatial predictors, ideally in the form of Moran Eigenvector Maps. Like R#1, I highly suggest you to use a partial RDA (or CCA) subdividing the dataset by variable type (lake morphology and limnological variables + space) and also repeat the analysis separately for each season. Or, if you want to explicitly incorporate the season in your analysis, consider using a STATICO analysis instead (see e.g., Ceron et al. 2020 Ecology and Evolution). Notice though that your ability to compare seasonal changes in community structure is very limited, because of small sample size.
If you want to compare species richness between seasons (your results) and among regions (comparing to the literature) you MUST use rarefaction, see methods of Colwell et al. 2012 J. Plant Ecol, Chao et al. 2014 and Ecol Monog 2020 Chao et al. 2020 Ecol Res. implemented in the R package iNExT.

You also have to better explain why you measured those specific limnological variables.

A CCA doesn't test for a difference in species composition between sites, and they actually don't look separated in the diagram either.
If you want to test for a difference in species composition between seasons, use a JSDM implemented in mvabund::manyglm package

Overall, the paper looks very preliminary. As a consequence, the discussion is quite shallow. See also reviewers' comments on this. Stating an explicit hypothesis at the end of the introduction would help readers to understand why you used the statistical methods you presented.

Minor comments:

1) Table 1 and 2 can be combined since they show the same information, just put an additional row to separate data from dry and wet season.

Remember to include in your Rebuttal letter all the comments from reviewers and mine, including those in the PDF attached.

Reviewer 1 ·

Basic reporting

The manuscript is very straightforward and clearly shows the high diversity of zooplankton in a set of small waterbodies in Southwest China. Apart from this important description, comparing the magnitude of zooplankton diversity with other much larger localities (but with similar or lower diversity!), authors also explored differences among seasons and the relationships between local species compositions and environmental variables with canonical correspondence analysis. I believe this last goal deserve a little more attention. The correlation between species composition and environmental variables assume that environment is filtering the species able to occur in a certain site. Authors selected a set of limnological variables as candidates of likely predictors of species abundance, which is fine. It seems most variables indicate a gradient of productivity. It is unclear how collinearity was treated, in the first place. Apart from that, I believe that authors could also include other candidate variables related to major waterbody features, maybe authors can dig a little further and get depth, area and margin perimeter of the waterbodies (maybe others). This other set of predictors relates to waterbody morphology, which can also be an explanation of species sorting among sites. Moreover, the chance of dispersal among sites can also explain species abundance differences, as described in metacommunity framework (see a review in Heino et al. 2015, Freshwater Biology). Then, dispersal routes among waterbodies can be represented by spatial variables generated using Principal Coordinate of Neighbour Matrices (PCNM) to generate spatial predictors. Therefore, you can use partial redundancy analyses (pRDA) instead of CCA to explore relationship between species abundance and predictors, using three predictor matrices (limnological, morphological and spatial variables). This analysis will explore the relative role of likely different mechanisms explaining species abundance variation. Also, to avoid overfitting and collinearity, you may use a forward selection in variables of matrices before pRDA. Given the seasonality observed in sampling sites, this analysis could be done for each season separately, which could be used to explore of relative roles of different mechanisms are seasonal. Finally, I would suggest you to avoid interpretations on causality, but focus on interpretations of which mechanisms correlates indicates. Apart from this general comment, I must congratulate authors for text fluency and clarity, and also for the amazing report. Considering this and the fact that I am not also a native speaker, I preferred not to indicate comments of the text to preserve author’s writing style.

Experimental design

Comments in topic 1.

Validity of the findings

Comments in topic 1.

Additional comments

Comments in topic 1.

Reviewer 2 ·

Basic reporting

The English language is overall well-written, but the text would benefit with many (lacking) details provided in Introduction, Methods and Results as I exemplify below. Authors have a great amount of data that can be better explored and displayed in the text. Tables and figures might also be changed in order to better display the obtained results. I believe the field sampling and lab routine is correctly performed, but in the present form the results are under-evaluated. Many analyses on basic community ecology might be performed to better display the results and reach the goals of the study. As a consequence of such changes the Discussion will benefit.

In the Introduction, the text would benefit if small wetlands are contextualized in an ecological perspective, displaying their importance for aquatic organisms, their temporary nature (if this is the case in this study), isolation from other aquatic systems, lack of fish, threats and conservation value. In the current form the text shows explicitly the context of wetlands in China. Especially for microcrustaceans small wetlands are valuable because they usually lack fish and display high habitat heterogeneity mediated by aquatic macrophytes. The text would reach more general ecologists if such context is present in the text. In addition, the text states for spatiotemporal variations in crustaceans, but note that no spatial variables are included in the text (such as coordinates) (see lines 31, 70, 136). I suggest including spatial variables or removing spatial variables from the text. Finally, based on previous knowledge about wetlands and other aquatic systems in the region, can authors provide any hypothesis that was tested?

Experimental design

Lines 73 – 81 – there is some division of the landscape in west and east sides plus upstream and downstream subgroups, but note that this separation are not important for data analyses, results or discussion. Consider removing them.

In figure 1, consider showing the study site also in the context of China. I believe most readers are not aware of Yunnan location.

Please provide more details on the aquatic systems: do they have fish? How about aquatic macrophytes? Are they permanent or temporary? (I suppose some are temporary). What affects their hydroregime? (rains, hydroconnectivity to other systems, watertable changes). How often they connect to each other? What is their average depth or range? All these information may help understanding the patterns of crustacean communities. Note that this study observed high species richness when compared to others. Maybe the lack of fish predation may partially explain this pattern? See the paper from Scheffer et al. (2006) in Oikos. “Small habitat size and isolation can promote species richness: second‐order effects on biodiversity in shallow lakes and ponds.”
Line 93 – How 20L of water were collected? By buckets? In open water or macrophytes stands?

Statistical analysis lacks many tests displayed later in the results and may benefit also from commonly used methods in community ecology. Briefly, please indicate that Mann-whitney tests were used to compare species richness and abundance. Even better, consider using rarefaction curves or rarefied species richness to compare species richness between rainy and dry seasons. Community composition may be compared with PerManova and a non-metric multidimensional scaling will nicely display community variation. SIMPLER or Indval may be used to find species that are typical of rainy or dry seasons. Note that in the current form many results are descriptive.

Consider displaying exact p values instead of < 0.05 or > 0.05 (see lines 130 and 135, for example)

Validity of the findings

Please write the full genus name of each species when first mentioning them.

Consider using the above mentioned methods to compare community metrics: rarefaction curves, NMDS, PerManova, SIMPLER, IndVal.

In tables 1 and 2, what is meant by % of area? Is that % of lakes? Please clarify. In addition, consider separating species by cladocerans and copepods.

Figure 2A – The axis is a bit weird as it varies from 5 to 1 to 5. Please double check it.

Before CCA, can authors provide a brief description of limnological variables? Raw data is provided as supplemental material, but I believe the text will benefit with a table showing maximum, minimum and mean values for each season. For CCA, were limnological variables previously correlated between them to avoid covariation?


Discussion

The structure of Discussion might be changed if more analyses are provided in the manuscript. I suggest that each community parameter (species richness, abundance, composition) can be discussed taking into account the seasons (rainy and dry). The text might also benefit from some implication of the findings, provided in the conclusions. I provide below comments on specific topics.

Lines 165-167 – Use Permanova to state that communities differ. I believe the use of % of occurrence is not the most appropriate method.

Lines 180-182 – consider discussing the role of fish predation. See the paper from Scheffer et al. (2006) in Oikos mentioned above. Small habitat size and isolation can promote species richness: second‐order effects on biodiversity in shallow lakes and ponds.

Line 186 – I am not sure that in your study area nitrogen increased algae biomass, which in turn supported crustaceans. If that happened, why algae biomass did not differ between rainy and dry seasons? Please show chlorophyll a content results.

Line 203 – Dissolved oxygen values are very high. Please double check it.

With more analyses, specific to each community parameter, the Discussion can be rephrased. I suggest contrasting rainy and dry seasons whenever possible.

---

## Round 0.2 · Minor Revisions

One of the previous reviewer has commented on this revised version. Overall, the manuscript has improved considerably. Thanks for incorporating all the suggestions from the previous round of review.
I do have a couple more suggestions, though. Please, see PDF attached.

Reviewer 2 ·

Basic reporting

I appreciate the effort made in the current version of the ms as most of the previous suggestions were taken into account. The text is now easier to read and more straightforward in my view. However, I would suggest a few items to improve the final version:

Experimental design

No comment

Validity of the findings

No comment

Additional comments

Species diversity and community structure of crustacean zooplankton in the highland small waterbodies in Northwest Yunnan, China
by Chen et al.

I appreciate the effort made in the current version of the ms as most of the previous suggestions were taken into account. The text is now easier to read and more straightforward in my view.

I would suggest a few items to improve the final version:

1) Is it possible to infer what caused the differences in crustaceans between the dry and rainy seasons? Maybe the ponds are more variable in terms of limnological variables in the dry season as they receive less water, allowing more species to colonize the water column?

2) Some results on RDA are displayed in the Discussion for the first time, which is not usual. See line 235 stating that "We identified the limnological variables could explain the most variation of crustacean zooplankton community in both the rainy (NO3N, DSi, Cond, DO) and dry (NO and WT) seasons. Consider displaying the information on RDA in the Results section.

3) Finally, I see many issues in English language, especially in the Discussion section. I suggest reviewing all the text. Some examples follow:

lines 80-83 - goal 3 - Something is missing in this sentence.
lines 114 - 115 - Something is missing in this sentence.
line 152 - to reduce the weight of the ABUNDANT species....
line 211 - complicated habitats? Do you mean heterogeneous habitat?

---

## Round 0.3 · accepted · Accept

I'm satisfied with your latest responses and I'm glad to recommend acceptance of the paper as is. Congratulations.